# Enterprising Self through Higher Education: Youth, Aspirations and Future Amidst Academic Entrepreneurialism

Hei-Hang Hayes Tang [1,*] and Yan Zhang [2]

1   Department of Education Policy and Leadership, The Education University of Hong Kong,
    Hong Kong 999007, China
2   Institute of Education, University College London, London WC1E 6BT, UK
*   Correspondence: hhhtang@eduhk.hk

**Abstract:** This paper examines the macro trends, policy responses, and their impact on youth and their aspirations about the future. It uses the conceptualisation of "the self as enterprise" to focus our discussions about how, and to what extent, youth try to "enterprise" their selves through higher education amidst the global rise of academic entrepreneurialism. The paper addresses the new reality of changing economies and creative disruptions of employment markets facing youth and their futures. We will talk about the way youth career aspirations will be reconsidered and the role of higher education in it. We also discuss the in-betweenness of materialistic and post-materialistic pursuits among young people, as well as the topic about "enterprising self" through higher education and innovating futures using youth's entrepreneurial mindset, competence, and "self-entrepreneurship". The paper will end by discussing some inspirations and insights for policies and practices and empirical research concerning the topic of youth, futures, and aspirations.

**Keywords:** higher education; youth aspirations; future; academic entrepreneurialism





## 1. Introduction

In his book, *Sociology in Question*, Pierre Bourdieu famously claimed that "youth is just a word" [1]. The notion of youth has been evolving into a social construction and is instrumental for national governments to utilise as a social technology for both celebration and control [2]. The paradoxical construction portrays youth as the source of problems current in the society and, at the same time, as the polity's future hopes. Being a site of moral crises and the societal gaze, youth draw people's attention while ignoring the deeper underlying problems and contradictions in the neoliberal configurations of many advanced countries [3]. Different discourses about the young generation are shaped by contesting social and political forces, which turn "youth" into a fluid concept and complex signifier that is subject to alter both vertically (by defining an age range) and horizontally (social class, race/ethnicity, gender).

Youth is a common social issue for the concerns of social justice, equality, and equity. Young people from underprivileged socioeconomic circumstances in general lack resources and are poor in future aspirations, struggling to connect with school life and find purpose in a society that tends to marginalise their status. They fall under the social categories of "disengaged youth", "youth at risk", or other similar social labels. They are seen as the social problem in need of "fixing" [4] and have become the target of official interventions that may be punitive, therapeutic, or medical [5]. They lack the capacity to aspire for the ever-changing future and hence are less "future-literate" [6–9] than their counterparts from better-off families.

This conceptual paper offers a brief and succinct review about the macro trends, policy responses, and their impact on youth and their aspirations about the future. It uses the conceptualisation of "the self as enterprise" [10,11] to focus our discussions about how,

and to what extent, youth try to "enterprise" their selves through higher education amidst the global rise of academic entrepreneurialism [12–15]. The remaining part of the paper will be followed by Section 2, which addresses the new reality of changing economies and the way youth career aspirations will be reconsidered amidst the in-betweenness of materialistic and post-materialistic pursuits among young people. Section 3 addresses the topic about "enterprising self" through higher education and innovating futures using youth's entrepreneurial mindset, competence, and "self-entrepreneurship". Finally, Section 4 discusses some inspirations and insights for policies, practices, and recommendations for future empirical research concerning the topics of youth aspirations, futures, and higher education.

## 2. Changing Economies and Creative Disruptions of Employment Markets: Between Materialistic and Post-Materialistic Pursuits

In understanding the socio-political underpinnings of the discourse of youth, it is important to examine the relationship between labour, capital, and the role of higher education amidst globalisation of technological development and academic entrepreneurialism. Creatively disrupting the structure of economic and employment systems, digital transformation re-defines the skills demanded in the workforce as well as organisation and knowledge contents in higher education programmes. With the development of automation and artificial intelligence, low-skilled individuals—and medium-skilled ones later—may be replaced by machines. Such structural changes alter the types, diversity, and scope of realistic ambitions available for youth [16]. The new generation of youth need new mindsets, skills, and entrepreneurial capacities to navigate the opportunities in the changing economic and employment systems. On the other hand, the overt challenges, difficulties, and uncertainties encountered by youth in their life engagement may lead to frustration and drive them into being inactive [17]. There are tremendous demands on higher education to respond to so as to equip and empower young people for a precarious future.

In the digital age, young people are starting their work life, career, or other forms of life engagement in economic and employment structures with substantial precarity and risks [18]. Universities and colleges are expected to play the social role of empowering young people in the face of the changing future through higher education and disciplinary training of a diversity of fields and domains. To a certain extent, young people choose their educational and professional paths based on their career aspirations and vice versa. Meanwhile, career aspirations are shaped by social contexts through personal direct experience or second-hand experiences gained from friends, parents, teachers, or role models [19,20]. Given the uncertainty and complexity in the globalising and changing economic and employment structures, the disparity between the labour market situation and career goals is, however, expanding. Declining graduate employment has been reported internationally, especially in advanced countries with high participation in higher education [21]. In a situation where realistic aspirations are rare, people, however, are motivated to pursue education and skill development that can open up more views, broader perspectives, and inspire new future aspirations [16]. While young people continue to view a good wage, career advancement prospects, and social security as essential components of the ideal job, they are also becoming more open to increased workplace flexibility and a healthy work–life balance.

Across different perspectives on how new technologies may impact the economy, workforce, and opportunity structure, there are mainly two types. One demonstrates profound concerns, worry, and pessimism that jobs will be displaced by the learning capacity and performance of machines and artificial intelligence. The other projects a more general technical optimism that new technologies will lead digital transformations of many kinds but also create new jobs. What inspires optimism for young people is that they have been exposed to the trend of ever-updating technologies at an earlier age than previous generations have. Youth, especially with the impact of higher education, tend to be more

confident, skilled, and resilient with the technological changes and transformations they bring along [16].

Regarding the formation of their future aspirations, young people across the world reported that life purpose, love, family, friendship, healthy work–life balance, flexibility, job satisfaction, and social protection are more significant than money or worldly positions. According to the research by Hoskins and Barker (2017) [22], while wealth and status are essential to certain young participants, the study discovered that few of the participants reported a desire to move across the occupational ladder and nearly no one discussed upward mobility from the socioeconomic status of their parents. Very few of them even indicated that they were unhappy with their family socioeconomic background or way of life. Young people's aspirations for the future are overwhelmingly influenced by family-derived habitus and dispositions constructed through early childhood socialisation and family backgrounds, norms, and values.

Where there is evidence of a desire for money, it typically stems from a need to assure their financial security in the future, a drive rooted in past experiences with financial difficulties [22]. Boateng and Löwe (2018) [23] found that it makes sense to work and earn savings for raising their family as a medium goal. Moreover, a decent income can help them earn savings and improve their family's well-being. However, in the long term they desire an occupation or profession that is meaningful and requires less physical labour. After financial needs are fulfilled, interest and demands that are more intrinsic to life goals and meaning can start to emerge.

Adopting an international perspective, among countries with lower human development index, their young people tend to see salary as an important criterion when considering employment opportunities. In the meantime, youth in countries with higher levels of human development have less intention to go abroad for opportunities. From a continental perspective, young people in Europe and Central Asia do not consider salary and financial recompense the most crucial factor when choosing a job. More young women than men rank a work–life balance, sense of purpose, and contribution to society, compared with wages and benefits, as more significant factors for deciding what occupation to select. While both female and male youth in Europe and Asia are concerned about excessive competition and a lack of working experience when navigating their futures, European women are more likely than their male counterparts to raise concerns about the shortage of job opportunities and discrimination by employers [16].

Responding to the creative disruption of employment structure by the advancement of innovative technology and the deep impact of neoliberalism, educated youth are rethinking the meaning of work, leisure, and good life. They are, in the meantime, reconsidering the relationship between self and future. Higher education plays an unprecedented role of engaging the young in the view of their future and aspiration formation for the age of entrepreneurialism. We will discuss these issues in the next section.

## 3. Enterprising Self through Higher Education, Innovating Futures by "Self-Entrepreneurship"

At the advent of the "entrepreneurial turn" of higher education and academic professions [24], university education has been utilised as an imperative social technology that guides and "empowers" young people to develop their individual self and innovate their future as an "enterprise". The conception of "the self as enterprise" is an individualised and economic notion [25] reconfiguring other understandings about the self as a significant member of a citizenry and a constituent of collective wellbeing and bonding. Highly embedded in the neoliberal governance, political economy, and social systems, individuals are socialised and educated, or otherwise penalised, to be responsible for self-management and future-making and for being an entrepreneur of their own life and life chances. Humans are autonomous in planning strategically, investing wisely, excelling effortlessly, and accumulating individual capitals for their life and future. People relate to their own self and subjectivities through self-regulation and discipline and to others as competitors and

subjects of comparison. Incompetent self-management, low capabilities of self-discipline, and uneducated life planning and strategies are causes of poverty. The neoliberal idea of self-entrepreneurship [10] erodes the public good of civic engagement and care as it atomises social relations across social classes in society [11]. Foucault (2008) [25] puts forward the notion of "homo œconomicus" to present the conception of "the self as enterprise":

> Homo œconomicus is an entrepreneur: an entrepreneur of himself. This is true to the extent that, in practice, the stake in all neoliberal analyses is the replacement every time of homo œconomicus as partner of exchange with a homo œconomicus as entrepreneur of himself, being for himself his own capital, being for himself his own producer, being for himself the source of [his] earnings. (p. 226)

McNay (2009) [11] furthers and argues that central to self-entrepreneurship is an ontology that absolutises a total individualised notion of economic interest and marginalises other notions of the citizen-subject. She articulates:

> the organization of society around a multiplicity of individual enterprises profoundly depoliticizes social and political relations by fragmenting collective values of care, duty, and obligation, and displacing them back on to the managed autonomy of the individual. (p. 65)

For individuals to innovate different possibilities of the future, knowledge, learning capacity, and soft skills are altogether as they are essential for experimenting novel ideas, forging new paths, generating new opportunities, and starting entrepreneurship or new industries that the world never imagined. The engagement of university graduates has become a key issue in the age of the post-massification of higher education. "Raising" aspirations among under-represented groups [26] serves not only as a social inclusion strategy that widens university participation but ultimately as a strategy to increase national competitiveness in the global knowledge economy [27]. In particular, the employability and entrepreneurial competence of graduates are the new concerns of policies and practices of higher education systems across the world. Young people are exhorted to learn to become "economic citizens" [28] embodying agility, mobility, and entrepreneurial mindset [29]. The literature of youth aspirations and entrepreneurship reveals the imperative of soft skill development when young people aspire or "enterprise" themselves for their future. Although young people usually overlook or do not consider an entrepreneurial pathway as one of high priority [16], the new reality of the COVID-19 global pandemic creatively disrupted the precarious employment market and digitally transformed patterns of work, employment, and life engagement. Universities are needed for a future-oriented higher education that enables young people to master the mindset, knowledge, skills, and literacy to imagine and navigate their future.

According to anthropologist Arjun Appadurai (2013) [30], humans are "future-makers" (p. 285) and the "conceptions both about the future and the past, are embedded and nourished in culture" (p. 179). Compared with their peers who experience disadvantage and poverty, youth from more privileged families are exposed to a wider range of immediate opportunities and material goods, better positioned to explore and benefit from a variety of experiences of exploration and trial, and relate them to the concrete formulation of aspiration, option, and strategic planning. Advantaged young people embody greater cultural and cognitive resources to make sense, envisage, and communicate their aspiration and "road map" to their futures. They tend to have wider opportunities in their everyday life to cultivate a "futures literacy" and connect their future aspirations with larger social and global scenes and situations, as well as to even more abstract beliefs, narratives, and standards. The ability to aspire, as a result, is a navigational capacity [30]. Youth from well off socio-economic backgrounds possess great cultural capital, social networks, and informational resources to navigate their future more relevantly and realistically. Compared with their less privileged peers, they utilise "the map of the norms" of the aspiring future to develop their future literacy and communicate frequently with people in their social circles. Aspirational capacity is cultural in a sense that it derives its power from dominant systems

of value, meaning, interaction, and discourses about what the future should mean. Such a biased reality makes less resourceful youth possess less developed capacities to realise their aspirations [31] and make it more difficult to develop optimism about the future.

Capital, be it in the form of economic, social, or cultural, enables advantaged youth to utilise and take better advantages of postmodern liquidity [32] and choice [5]. In the current and future age of globalisation, international mobility is a means to upward socio-economic mobility. Youth from privileged families are more likely to take advantage of the benefits of having access to international networks and the ability to move around the world. Hope can be considered as a form of capital [33], whereas social class matters for the possession and disposition, or habitus, of hope and aspirations. Therefore, there is a clear and timely demand of social justice for universities and postsecondary institutions in youth empowerment amidst higher education massification. This paper concludes by recommending some inspirations and insights for policies and practices, especially the positive impacts that higher education can produce through nurturing the future literacy of youth from different social classes.

## 4. Inspirations and Insights for Policies, Practices, and Empirical Research

As illustrated above, the literature found that some young people do not take social mobility for granted but prefer maintaining familial social reproduction and sustainable well-being [22]. Given the gap and misalignment between youth's actual aspirations and the neo-liberal focus on enterprising self, we recommend policy makers and practitioners of higher education and youth policies to accommodate diverse, democratic, and innovative worldviews to consistently update and reframe policy frameworks that can deliver impact of youth empowerment for their futures. Entering the age of academic entrepreneurialism, wider access to postsecondary education materialises alongside neoliberal policies of study loans. Amidst the higher participation in higher education [34,35], it is extensively reported that disadvantaged students from low socioeconomic backgrounds usually leave colleges and universities with loans that require years to repay. Moreover, an oversupply of graduates does not guarantee an economic return of a university degree because education-occupation misalignment is common and it is a reason for universities not to see producing "job-ready" graduates as part of their mission.

We should reconsider the appropriateness and actual outcomes of policies intending to promote intragenerational social mobility by shaping youth to aspire highly through education-led programmes. Research found that family socioeconomic status is a key factor of forming and sustaining high aspirations throughout formal schooling processes [36]. Culturally speaking, policymakers need to be sensitive whether their aspirational programmes and interventions are indoctrinating the neoliberal middle-class values and "othering" towards the working class youth [22]. The disconnection between aspirations of youth from disadvantaged families and "optimism projects" by policy interventions may create emotional risk [26]. Youth voices and differences of opinion should be heard in order to ask the questions that make sense to the young people whom policymakers intend to understand [16]. The sociological factors of class, ethnicity, and social context affect the process of how youth aspirations are formed [37]. Additionally, higher education systems and young people are situated in their specific circumstances across advanced nations and developing nations, as well as post-industrial economies and rising industrial economies. Youth at risk should not be labelled as "disengaged" but considered "disenfranchised". Trying to "fix" young people does not help reduce the social inequity and inequality that is created by unchecked neo-liberal configurations of capitalism [5].

Based on the succinct conceptual review, we would like to recommend these topics and areas for future empirical research:

- What is "future" and what does it mean to youth from diverse backgrounds (national contexts, educational qualifications, nationalities, ethnicities, social class, gender)? What are the patterns in diverse youth voices about aspirations, futures, and higher education?

- How are aspirations formed and shaped and what are the future-crafting processes across the lifespan of teenage years and young adulthood?
- Considering education as an intervention for youth aspiration formation, what is the relationship between study major (hard vs. soft disciplines; applied/professional vs. basic/generic majors), university experiences, and self-enterprising projects through higher education?

### 5. Conclusions

This paper reports empirical highlights from youth research that consistently discovered that youth aspire and desire to be happy, make a difference, and experience satisfaction after engagement rather than enterprising their selves for social mobility and wealth creation through higher education. This indicates a perspective and reasoning different from dominant entrepreneurial discourses embedded in global capitalism and neoliberalism. The global massification of higher education extends and intensifies the impact of formal education beyond the life stage of adolescence until young adulthood. Educated youth mature and acquire higher aspirations. In particular, they are exposed to post-materialistic ideas and ideals, as well as develop a critical thinking about the negotiated choices between materialistic and post-materialistic pursuits, the differences between the means and ends of ultimate life goals.

Youth aspirations are malleable. For building a socially just political economy, we should not only be concerned with the structural circumstances of youth, but also innovate conceptual frameworks that theorise youth studies and policy research about youth across social class and cultural backgrounds. As for now, there are space and new possibilities for enhancement in terms of research paradigms, methodologies, and methods. With a view to creating meaningful input for policy-making and programmatic innovations for youth engagement and empowerment, global society needs a ground-breaking knowledge framework that critically examines and understands the realities of the political economy in which neoliberalism inexorably shapes the concept of youth, regardless of whether the rhetoric is stigmatising youth or being perceived as a positive means of change. Youth educational and employment aspirations are more likely to be achieved when policies align meaningfully with the contextualised pathways to realising them. It appears to be promising and productive to problematise the ideological underpinnings of youth discourse and questing for inter-generational and multicultural dialogues that transcend the everyday instances of agency and resistance. An unquestioned self-enterprising project through higher education, which is increasingly embedding itself in academic entrepreneurialism, does not foster genuine youth aspirations and optimism for the future.

**Author Contributions:** Conceptualization, H.-H.H.T.; writing—original draft preparation, H.-H.H.T. and Y.Z.; writing—review and editing, H.-H.H.T. and Y.Z. All authors have read and agreed to the published version of the manuscript.

**Funding:** This research received no external funding.

**Institutional Review Board Statement:** Not applicable.

**Informed Consent Statement:** Not applicable.

**Data Availability Statement:** Not applicable.

**Conflicts of Interest:** The authors declare no conflict of interest.

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
