# Peer review of "Enterprising Self through Higher Education: Youth, Aspirations and Future Amidst Academic Entrepreneurialism"

_2673-995X, doi:10.3390/youth2040045_

Round 1
Reviewer 1 Report
The introduction presents a clear first two paragraphs. At line 39-40 the paper moves from talking about the social construction of young people and the youth question/ youth problem to describing changes in labour and capital. While aligned, the movement between topics requires more connection. I would continue with a focus on young people and then move into the discussion between labour and capital to assist with the transition from the first two paragraphs to the third.
Section two describes the disruptions due to technologies and some career but does not fully articulate the disruption around higher education, thereby not fully connecting market disruptions to youth attitudes to higher education.
Section three presents the youth perspectives on life and what some call leading a good life. The section is clear about the shift from materialistic to post-materialistic. Some of the language in this section disrupts the reader (see line 106--"Another research by" can be omitted). Also the sentence starting on line 110 to end of paragraph can be written for better clarity of meaning.
Section 4, first sentence needs to be reworked for clarity. This section is one that the overall idea is not clear. There are many ideas presented in this section and not all are connected back to the idea of enterprising self through higher education. Many of the points seem to not relate this this overall idea at all, e.g. sentence that starts on line 146--Compared with their peers... I agree with the argument that the entrepreneurial emphasis within higher education continues a neoliberalism mission which misses (maybe ignores) much of the socio-political structures in which young people have to navigate, yet this is not clearly stated in the section
Author Response
(Please find the reply in the attached file)

Reviewer 2 Report
State a more specific conclusion in the abstract
The paper is very ambitious with all the topics it seeks to address and it does not cover them all, it is better to delimit these aspects from the summary and the introduction
It is important to propose a methodology section where the questions that are formulated in the article are clear (list them) and in this way give clarity about the methods to arrive at the proposed answers.
The results must respond to the approaches of the methodology. In addition, the results should account for the entrepreneurial difference between young people from developed countries and young people from developing countries.
The conclusions section should not carry references.
It is suggested to propose a section of the research agenda or future research that emerges from the reported findings.
Author Response

(The authors gave the same response as above.)

Round 2
Reviewer 2 Report
The authors have adequately responded to each request